# Kynurenine Metabolism as a Mechanism to Improve Fatigue and Physical Function in Postmenopausal Breast Cancer Survivors Following Resistance Training

**DOI:** 10.3390/jfmk7020045

**Published:** 2022-06-01

**Authors:** Ronna N. Robbins, Jessica L. Kelleher, Priyathama Vellanki, Jason C. O’Connor, Jennifer S. Mascaro, Joe R. Nocera, Monica C. Serra

**Affiliations:** 1South Texas Veterans Health Care System, San Antonio, TX 78229, USA; oconnorj@uthscsa.edu (J.C.O.); serram@uthscsa.edu (M.C.S.); 2Atlanta VA Health Care System, Atlanta, GA 30308, USA; jessica.kelleher@va.gov (J.L.K.); joenocera@emory.edu (J.R.N.); 3Division of Endocrinology, Department of Medicine, Emory University School of Medicine, Atlanta, GA 30322, USA; priyathama.vellanki@emory.edu; 4Department of Pharmacology, UT Health San Antonio, San Antonio, TX 78229, USA; 5Division of Preventive Medicine, Department of Family and Preventive Medicine, Emory University School of Medicine, Atlanta, GA 30322, USA; jmascar@emory.edu; 6Department of Rehabilitation Medicine, Emory University School of Medicine, Atlanta, GA 30322, USA; 7Division of Geriatrics, Gerontology & Palliative Medicine and the Sam & Ann Barshop Institute for Longevity & Aging, UT Health San Antonio, San Antonio, TX 78229, USA

**Keywords:** kynurenine, fatigue, breast cancer survivors, resistance training, cognitively based compassion training

## Abstract

This pilot examines whether resistance training (RT) can induce changes in kynurenine (KYN) metabolism, which may contribute to improved physical function in breast cancer survivors (BCSs). Thirty-six BCSs (63.2 ± 1.1 years) underwent assessments of physical function and visual analog scale (100 cm) fatigue and quality of life before and after 12 weeks of RT (N = 22) or non-exercise control (CBCT©: Cognitively Based Compassion Training, N = 10). Blood was collected before and after interventions for assessment of KYN, kynurenic acid (KYNA), and peroxisome proliferator-activated receptor γ co-activator 1α (PGC-1α). At baseline, the women were moderately fatigued (mean score: 46 cm) and at risk of poor functional mobility. A group*time interaction was observed for all measures of strength with improvements (~25–35%) following RT (*p*’s < 0.01), but not CBCT. Time effects were observed for fatigue (−36%) and quality of life (5%) (*p*’s < 0.01), where both groups improved in a similar manner. A group*time interaction was observed for KYN (*p* = 0.02) and PGC-1α (*p* < 0.05), with KYN decreasing and PGC-1α increasing following RT and the opposite following CBCT. These changes resulted in KYN/KYNA decreasing 34% post-RT, but increasing 21% following CBCT. These data support RT as a therapeutic intervention to counteract the long-term side effect of fatigue and physical dysfunction in BCSs. Additionally, the results suggest that this effect may be mediated through the activation of PGC-1α leading to alterations in KYN metabolism.

## 1. Introduction

The number of cancer survivors in the U.S. continues to grow, with 16.9 million survivors in January 2019, which is projected to increase to 22.1 million by 2030. This projected increase is secondary to advances in early cancer detection and treatment, increased longevity, and the growth of an aging population [1]. Breast cancer is one of the three most prevalent cancers in the U.S., with over 250,000 patients newly diagnosed and over 3.8 million survivors in 2019 [1,2]. Despite a 10-year relative survival rate of 84%, breast cancer survivors (BCSs) often do not fully recover their premorbid behavioral and physical function [3], suggesting other factors affect recovery. With almost two-thirds of BCS aged 65 years and older [1], and older BCSs more likely to have multiple comorbidities compared to younger ones [4], it is imperative to understand the influence of cancer and its treatments on long-term recovery in older BCSs.

Approximately 90% of BCSs experience long-term sequalae secondary to their cancer treatment, severely impacting quality of life (QoL). Functional changes, including decreased aerobic capacity, strength, and mobility, along with physiological and cognitive dysfunction, pain, depression, anxiety, and fatigue are among the long-term side effects experienced by BCSs [5]. Compared to women who have not had breast cancer, 5-year postmenopausal BCSs are 40% more likely to report functional limitations (i.e., ability to perform intense household chores, walk half a mile, or climb stairs) [6]. Poorer lower-extremity muscle strength and less physical activity are associated with more severe fatigue and depression in BCSs [7,8], which can persist for years after completing cancer therapy [9]. Depression and cancer-related fatigue are two of the most prevalent side effects [9,10] and may endure for up to 10 years post-treatment, suggesting that treatments for psychological and physical dysfunctions remain poorly understood [11]. 

Increasing data suggest that chronic inflammation may be one of the mechanisms underlying cancer-related fatigue, depression, and mobility dysfunction in BCSs [12]. The latest literature also suggests that it is the inflammation-induced dysregulation of tryptophan degradation through the kynurenine (KYN) metabolic pathway that becomes altered during breast cancer treatment [13], which mediates behavioral and physical dysfunctions [14,15]. Therefore, interventions that reduce pro-inflammatory cytokines with the resultant stimulation of KYN metabolism may be increasingly important in BCSs to prevent susceptibility to depression, cancer-related fatigue, and declines in muscle mass and function. 

The anti-inflammatory properties and their influence on the KYN pathway make exercise a potential therapy in the management of adverse side effects in BCSs [16,17]. The evidence supports exercise as a non-pharmacological therapy to counteract the progression of psychological disorders, including depression, fatigue, sedentary behaviors, and obesity, during cancer survivorship [18]. Among the various types of exercise interventions, resistance training (RT) is emerging as one of the most efficacious at reducing fatigue in BCSs [19]. We have shown that RT reduces fatigue and improves muscle mass and physical function in BCSs, which may be impacted by changes in inflammatory cytokines [20]. Recent evidence in human and animal models suggests that exercise-induced changes in KYN metabolism are mediated by the peroxisome proliferator-activated receptor γ co-activator 1α (PGC-1α), inducing a shift of KYN to kynurenic acid (KYNA), which may protect individuals from depression and reduce cancer-related fatigue [21]. This is further supported by the evidence from skeletal-muscle-specific PGC-1α knock-out animals, which have reduced physical functioning and elevated markers of inflammation [22]. Considering the anti-inflammatory properties of exercise along with the influence of the skeletal muscles on the KYN pathway [16,17], this study aims to examine the effects of RT on KYN metabolism and inflammation as a mechanism to promote changes in behavioral and physical functions in postmenopausal BCSs. These findings may contribute to the literature by identify potential targets for interventions to promote cancer recovery.

## 2. Materials and Methods

### 2.1. Recruitment and Screening

Eighty-six women were recruited for this study through local medical advertisements. Institutional Review Board-approved written informed consent was obtained from 43 potential women. Women (50–73 years of age, no menses ≥ 1 year) within 6 months to 10 years post-active breast cancer therapy (i.e., surgery or chemotherapy) for stages I–III invasive breast cancer, at least 6 months post-breast reconstruction (if applicable), untrained with regard to structured RT (no more than 3×/week), and having a BMI 20–45 kg/m^2^ and self-reported fatigue (≥3 on a 1–10 scale), were enrolled. Women with plans for surgery (e.g., breast reconstruction) during the study period, an orthopedic or chronic pain condition restricting exercise, Mini-Mental Status Examination (MMSE) scores below education-specific cut-points (less than 23 for higher than 9th grade education and less than 17 for lower than 8th grade education), or who were unable to receive physician medical clearance were excluded from the study. Of the 43 women who signed the consent, 7 were found to be medically ineligible and 36 were enrolled. Medical clearance to participate was received from each woman’s primary care physician. All prescribed and over-the-counter medications and a medical history was self-reported by the women during an interview with the study staff.

### 2.2. Research Testing

Eligible women underwent the following research tests at baseline and after the 12-week interventions (described below).

#### 2.2.1. Anthropometric Measurements

Body weight and standing height were collected with subjects dressed in light clothing without shoes. Body Mass Index (BMI) was calculated as weight (kg)/height (m^2^) and then categorized into the Center for Disease Control Adult standard weight status categories: underweight: <18.5 kg/m^2^; healthy weight: 18.5–24.9 kg/m^2^; overweight: 25.0–29.9 kg/m^2^ and obese: ≥30.0 kg/m^2^ [23]. Using the standardized protocol [24], maximal waist circumference was measured at the greatest anterior extension of the abdomen, usually at the level of the umbilicus, with a Gulick (flexible tape that does not stretch) tape and the participant standing upright and relaxed.

#### 2.2.2. Mobility and Functional Test

Standardized instructions were provided by a trained research exercise physiologist for each test. Women completed the following functional assessments using standard guidelines: (a) 6 min walk distance (6MWD) [25], (b) 3 m timed up and go (TUG) [26] and (c) chair stands (time in secs to stand up and sit down 5 times) [27], using the fastest of two trials. Usual gait speed was assessed as the time it took the participant to walk a short distance on a level surface at their comfortable/natural walking speed. In addition to calculating mean scores, usual gait speed was compared with the cut-off value of <1 m/s associated with poor clinical and functional outcomes [28]. As cut-off values are not available for 6MWD, timed chair stands and the TUG, individual values were compared with age-matched normative values (normative range declining with age from 538 to 471 m) for 6MWD [26], ≤8 s for the TUG [26] and 11.4 to 12.6 s for 5 chair stands [27] to categorize each woman as at or above versus below the norm.

#### 2.2.3. Strength and Endurance Tests

Hand-grip strength of the dominate arm was assessed by handheld dynamometry. Measurements were taken in duplicate in order to obtain the highest of the two measurements. Individual values were compared with age-matched normative values (23.7 to 25.3 kg) to categorize each woman as at or above versus below the norm [29]. One repetition maximum (1 RM) strength that included 4–6 trials with set rest periods of the knee extension, leg press and chest press exercises was determined. 

### 2.3. Self-Report Measures

Visual analog scales (VASs) were used to measure the women’s experience of global pain, tiredness, fatigue and QoL. The VAS consists of a 100 mm horizontal line with each end anchored with the extremes of the symptoms (i.e., left end “no pain” and right end “worst possible pain”). Participants marked the line indicating the number of symptoms they felt at the current time. The score was determined by measuring the distance (mm) from the left end of the line to the participant’s mark. Scores for VASs ranged from 0–100 mm with a longer distance (mm) indicating a greater intensity of pain, tiredness and fatigue and better QoL. Mild, moderate and severe intensity of symptoms were classified as a score ≤ 34, 35 to 64 and ≥65, respectively [30].

### 2.4. Laboratory Analyses

Twenty ml of EDTA plasma was drawn following a 12 h fast and 24–48 h after the last RT and CBCT session. Plasma was processed according to the Standard Operating Procedures in Clinical Research [31]. Following collection, the blood was mixed by inverting the tube 8–10 times, and plasma was then immediately separated by centrifugation at 4 °C for 10 min at 1100× *g* and then stored at −80 °C until laboratory analyses. The following analyses for cardiometabolic health and inflammation and KYN metabolism were performed: Cardiometabolic Health: Metabolic syndrome (MetS) was defined following the American Heart Association National Heart, Lung, and Blood Institute (AHA/NHLBI (ATP III)) guidelines for the Americans as the presence of at least 3 of the following components: (1) elevated waist circumference (≥88 cm for women in the United States), (2) elevated triglycerides (TGs) (≥150 mg/dL) or drug treatment for elevated triglycerides, (3) low HDL cholesterol (<40 mg/dL for men and <50 mg/dL for women) or drug treatment for low HDL cholesterol, (4) elevated blood pressure (systolic ≥ 130 mm Hg, or diastolic ≥ 85 mm Hg, or both) or antihypertensive drug treatment for a history of hypertension, and (5) elevated fasting glucose (≥100 mg/dL) or drug treatment for elevated glucose [32]. Plasma TGs and cholesterol were analyzed by enzymatic methods (Hitachi model-917 analyzer), and high-density lipoprotein (HDL)-cholesterol measured in the supernatant following precipitation with dextran sulfate. Blood pressure was measured as the lowest of 2 consecutive resting measurements following 5 min of rest. Fasting plasma glucose and insulin were measured in duplicate on a Beckman AU480 chemistry analyzer (Brea, CA). Homeostatic Model Assessment of Insulin Resistance (HOMA-IR) was calculated as [(fasting insulin (µU/mL) × fasting glucose [mmol/L])/22.5] [33]. Leptin, adiponectin and resistin, which measure energy homeostasis and appetite regulation were measured in duplicate by commercially available enzyme-linked immunosorbent assays (ELISA; Boster Biological Technology, Pleasanton, CA, USA).Inflammation and Kynurenine Metabolism: Plasma high-sensitivity C-reactive protein (hs-CRP) was measured in duplicate on a Beckman AU480 chemistry analyzer. Plasma KYN (antibodies-online Inc., Limerick, PA, USA), KYNA (Abbexa, LLC, Houston, TX, USA) and PGC-1α (CUSABIO, Houston, TX, USA) were measured in duplicate by commercially available ELISA.

### 2.5. Interventions

Following baseline testing, women were randomly assigned (2:1 randomization) to the aRT group or to a non-exercise attention control (CBCT©: Cognitively Based Compassion Training) group for 12 weeks.

#### 2.5.1. Resistance Training

The facility-based 3×/week for 12 weeks RT protocol was led by an exercise physiologist and designed to provide a group-based high-volume, moderate-intensity whole-body training stimulus. Women performed 15 repetitions for two sets and to exhaustion on the third set for seven major muscle groups: the leg and chest press, knee extension, leg curl, row, abdominal crunch and bicep curl. Resistance was gradually increased to account for strength gains when the women were able to complete 20 repetitions on the third set. 

#### 2.5.2. Cognitively Based Compassion Training

The CBCT© served as the non-exercise attention control group and accounted for the effect of perceived social support on fatigue and physical discomfort in exercise [34]. This program was administered by a trained CBCT instructor who had fulfilled requirements for CBCT teacher certification through Emory University’s Center for Contemplative Science and Compassion-Based Ethics program. CBCT classes were performed in a group setting for 1.5–2 h per week for 12 weeks, and consisted of didactics, class discussions and guided meditation practice. Each class began with a period of meditation to calm and focus the mind, followed by analytical practices designed to challenge unexamined assumptions regarding feelings and actions toward others with a focus on generating spontaneous empathy and compassion for themselves and others. CBCT has been shown to be a feasible and highly satisfactory intervention that improves depression, well-being, mindlessness and vitality/fatigue in cancer survivors [35,36].

### 2.6. Statistical Analysis

Descriptive statistics were expressed as mean ± SEM or frequency and percent. Baseline values were compared using a Student’s *t*-test for continuous variables or Fisher’s exact test for categorical variables. Statistical analyses were performed using a 2-way ANOVA with RT and CBCT as in-between variables, and time as within variable. Because traditional post hoc testing in a repeated measures framework does not fully account for intra-individual variability, within-group changes were explored further using paired *t*-tests. These tests were two-tailed and *p* < 0.05 was considered statistically significant. Data were analyzed using SPSS (IBM Analytics, Armonk, NY, USA).

## 3. Results

### 3.1. Participant Characteristics

Of the 36 women who underwent baseline testing, one woman in the RT group and two women in the CBCT group withdrew due to time commitments, and another woman in the RT group withdrew due to lower-leg cramping deemed unrelated to study procedures. Therefore, data are reported for 32 women (N = 22 in RT group and N = 10 in CBCT group) who completed the intervention. On average, the RT group attended 86% and CBCT group attended 85% of available sessions, and all women attended at least 75% of sessions. 

Demographic data may be viewed in Table 1. At baseline, women were similar with regard to race (*p* = 0.14) and years post-active breast cancer diagnosis (*p* = 0.17); however, the mean age was ~6 years older in those in CBCT compared to those in RT (*p* = 0.02). Overall, the women were about five years post-active breast cancer diagnosis and 60% non-Hispanic White and 40% non-Hispanic African American. 

Data related to baseline function, self-report measures and cardiometabolic, inflammatory and KYN biomarkers may be viewed in Table 1. In general, the women had mean physical functions below normative or cut-point values, which was comparable between groups (6MWD: *p* = 0.96; TUG: *p* = 0.55; chair stands: *p* = 0.74; usual gait speed: *p* = 0.64, handgrip strength: *p* = 0.81). Furthermore, the self-report measures indicated that the women were moderately fatigued and mildly tired, but with good QoL (by VAS) at baseline, which were similar between groups (fatigue: *p* = 0.93; tiredness: *p* = 0.52; QoL: *p* = 0.53). However, baseline pain was 1.4 times higher in the RT vs. CBCT group (*p* = 0.03). Cardiometabolic (HOMA: *p* = 0.46; MetS: *p* = 0.48; adiponectin: *p* = 0.26; resistin: *p* = 0.42), inflammatory (hs-CRP: *p* = 0.86) and KYN biomarker concentrations also were similar between groups, except that concentrations of leptin and KYN were slightly higher in the RT vs. CBCT group (*p*’s <0.05). The majority of women had metabolic syndrome (58%), which was similar between groups (*p* = 0.16). 

### 3.2. Effect of RT vs. CBCT on Body Composition, Physical Functioning and Self-Reported Outcomes

Data related to the intervention effects may be viewed in Table 1. The weight of the women in both groups remained stable, indicated by no significant change in BMI during the respective interventions. As anticipated, a group*time interaction (*p*’s < 0.02) was observed for all measures of strength (knee extension, leg press and chest press) so that there were improvements (~25−35%) following RT (all *p*’s < 0.01), but not CBCT. There was a time effect for 6MWD and chair stands, so that both groups improved following the interventions (*p*’s < 0.05). Measures of function (TUG, gait speed and hand-grip strength) improved following RT, but these changes were not significantly different to those observed with CBCT. No group*time effects were observed for self-report measures. However, time effects were observed for fatigue (−36%) and QoL (5%) (*p*’s < 0.01), with both groups having similar improvements in these outcomes over the 12 weeks. Furthermore, a group effect was observed for pain (*p* = 0.03), so that pain started and remained elevated throughout the intervention in the RT groups, despite a ~37% decline in pain observed with RT.

### 3.3. Effect of RT vs. CBCT on Cardiometabolic Health

No significant changes were observed for cardiometabolic health (HOMA-IR, MetS, adiponectin and resistin). However, a time effect was observed for leptin (*p* < 0.05), so that leptin similarly increased over time in both groups (27% overall increase).

### 3.4. Effect of RT vs. CBCT on Kynurenine Metabolism

No changes in hs-CRP were observed. However, a group*time interaction was observed for KYN (*p* < 0.01) with concentrations decreasing by 29% following RT, but increasing 36% following CBCT. Conversely, a group*time interaction also was observed for PGC-1α (*p* < 0.05) with concentrations increasing by 27% following RT and decreasing 2% following CBCT. Furthermore, a trending time effect was observed for concentrations of KYNA (*p* = 0.07), which tended to increase over time. These changes resulted in a significant group*time interaction for KYN/KYNA (*p* = 0.02) with the ratio decreasing 34% post-RT, but increasing 21% following CBCT.

## 4. Discussion

This pilot trial was designed to examine the effects of RT on KYN metabolism as a mechanism to promote changes in behavioral and physical functions in postmenopausal BCSs. Our results add to the accumulating evidence [20,37,38] to support RT in BCSs as a safe, feasible, non-pharmacological modality to reduce fatigue and promote psychological and functional health.

Our findings also add to the literature by suggesting that KYN metabolism may be a mechanism for these improvements, as we observed a shift in KYN toward KYNA metabolism. Some of the most prevalent long-term side effects experienced by BCSs included fatigue and physical dysfunction that can persist for years following treatment [5,39]. Women in this study, who were years past active breast cancer, were at high risk for psychological, physical and metabolic dysfunctions at baseline. This contributes to the evidence suggesting that current healthcare systems cannot meet the long-term sequelae of BCSs [5]. As anticipated, women performing RT improved physical function, whereas no improvements in physical function were observed in the CBCT group. With women improving strength and function close to or above normative threshold values [26,27,29], RT had the potential to extend functional independence. Both RT and CBCT resulted in mean changes in fatigue that were considered clinically meaningful (7 points [40]), while RT also met the clinical cut-points for meaningfulness for mean improvements in pain (11 points [41]) and tiredness (7 points [40]).

Fatigue is multidimensional and can contribute to a decline in physical activity, function, QoL and increase in body weight [9]. It is shown that poor lower-extremity muscle strength is a significant predictor of fatigue in BCSs [7]. Even though the underlying mechanism of fatigue has not clearly been identified, exercise is widely accepted as a non-pharmacological intervention for promoting psychological and functional health [10,42]. Despite this, a recent study examining fatigue and exercise in BCS focus groups found that women noted the importance of exercise for weight control and cancer reoccurrence, but did not recognize its importance in managing fatigue [10]. Additionally, women had misconceptions regarding potential harm, which was a barrier to exercise, suggesting that education may be needed for exercise to be widely implemented as an effective therapy [10]. With accumulating evidence to support CBCT’s effectiveness to reduce fatigue, it could be utilized as an alternative therapy for women with barriers to exercise.

Our results do not support RT or CBCT as modalities to improve cardiometabolic health, as we did not observe changes in the number of MetS constituents, HOMA−IR or inflammation. With regard to exercise, this is refuted by prior studies that showed RT reduced the prevalence of MetS and inflammatory biomarkers in older women [43,44,45]. The discrepancies could be attributed to differences in RT modality, repetitions, duration and intensity, and the population of study (i.e., age, BMI and comorbidities). Additionally, with postmenopausal BCSs at increased risk of long-term functional limitations [6], the differences in baseline physical abilities between the study populations could also account for the variations in the study results. CBCT is also shown to reduce inflammatory profiles [46,47], but less is known about its effects on metabolic health. However, the evidence suggests that aerobic exercise and dietary modification resulting in 5–10% weight loss leads to metabolic improvements, which may result from declines in systemic KYN concentrations [48,49]. Therefore, the future research should consider the addition of other lifestyle interventions of RT and CBCT, such as aerobic exercise and dietary modification, which can simultaneously affect metabolic risk through the KYN pathway.

Preliminary data from this study suggest that the mechanism involved in changes in psychological and physical functions may differ between RT and CBCT. Under chronic pro-inflammatory conditions, tryptophan degradation via the KYN pathway is dysregulated, and evidence suggests that this dysregulation is responsible for many chronic diseases [16]. Recently, Zimmer et al. [50] investigated the effects of a 12-week supervised progressive RT on the KYN pathway in patients undergoing radiotherapy for breast cancer and showed a reduction in KYN following RT, but not a non-exercise control. Unfortunately, no clinical outcomes were reported by this group. Our preliminary data add to these findings by suggesting that RT-induced improvements in physical and psychological functions may be induced by changes in PGC-1α, which may mediate KYN/KYNA. Work on animal models suggests that this pathway maybe activated in skeletal muscles with exercise, as increases in skeletal muscle PGC1α results in greater glucose utilization, thus reducing skeletal muscle fatigue [51]. 

Our results should be interpreted in light of several limitations. Our study included a small sample size, which prevents controlling for potentially important baseline demographic differences between the RT and CBCT groups, since adding covariates to the analyses may lead to inflated type-1 error rates. However, only age differed between the groups at baseline, but the difference of ~6 years is not considered clinically meaningful in the context of diseases that are governed by similar age-related processes. Another limitation is that the study did not account for activities performed outside the intervention. Therefore, it is possible that the psychological effects of CBCT led to an increase in daily activity or other lifestyle changes (e.g., diet), which could have resulted in the improvement of fatigue and QoL. In this study, a significant decrease in plasma KYN was observed; however, there was no significant decrease in KYNA. With KYNA rapidly excreted by the kidneys after exercise, a urine analysis collected along with blood samples immediately post-exercise could help capture the greatest changes in KYNA concentrations. Furthermore, our current analyses were limited to blood, and with exercise upregulating muscle-derived transcription factor PGC−1α, which is responsible for the expression of enzymes that influence KYN metabolism, skeletal muscle biopsies could have provided tissue-level outcomes to assist with data interpretation. To accurately measure the changes in KYN metabolism, future studies should consider urine, blood and muscle samples measured at several time points to determine the acute and chronic effects of exercise.

## 5. Conclusions

In general, these findings support RT as a therapeutic intervention to counteract the long-term side effects of psychological and physical dysfunctions in BCSs. Our preliminary findings also suggest that this effect is mediated through the activation of PGC-1α, leading to alterations in KYN metabolism. For RT to be an effective therapy, education may be needed for BCSs who have misconceptions that exercise can cause potential harm. Future studies should also consider the combined effects of RT with other lifestyle interventions, including aerobic exercise and dietary modifications, as well as the tissue-specific mechanistic evaluation of KYN metabolism.

## Figures and Tables

**Table 1 jfmk-07-00045-t001:** Baseline characteristics and effects of RT on physical, psychological and metabolic functions.

	Resistance Training (N = 22)				Cognitively-Based Compassion Training (N = 10)						
	Pre	Post	Change	Change	Pre	Post	Change	Change	Overall Group	Overall Time	Group x Time
				*p*-Value				*p*-Value	*p*-Value	*p*-Value	*p*-Value
**Race**	**N, %**				**N, %**						
Caucasian	11, 50	-	-		8, 80	-	-	-	-	-	-
African American	11, 50	-	-		2, 20	-	-	-	-	-	-
	**Mean ± SEM**	**Mean ± SEM**	**Mean ± SEM**		**Mean ± SEM**	**Mean ± SEM**	**Mean ± SEM**				
**Age (years)**	61.6 ± 1.5	-	-	-	67.2 ± 1.2	-	-	-	-	-	-
**Post Active Breast Cancer** (months)	53.3 ± 8.7	-	-	-	74.1 ± 11.1	-	-	-	-	-	-
**BMI** (kg/m^2^)	32.7 ± 1.4	29.3 ± 2.2	−3.4 ± 2.2	0.14	28.9 ± 4.5	29.4 ± 1.5	0.5 ± 0.2	0.06	0.25	0.44	0.38
**Physical Functioning**											
6MWD (m)	466.4 ± 21.6	479.9 ± 20.7	13.6 ± 4.6	<0.01	468.3 ± 23.0	471.1 ± 22.2	2.8 ± 6.0	0.66	0.92	0.05	0.18
TUG (s)	7.0 ± 0.5	6.6 ± 0.9	−0.5 ± 0.3	0.12	6.6 ± 0.4	6.6 ± 0.4	0.0 ± 0.2	1.00	0.73	0.32	0.32
Usual Gait Speed (m/s)	1.2 ± 0.1	1.2 ± 0.1	0.0 ± 0.0	0.07	1.1 ± 0.1	1.1 ± 0.2	0.0 ± 0.0	0.90	0.67	0.87	0.76
Chair Stands	10.9 ± 0.6	9.5 ± 0.6	−1.3 ± 0.0	<0.01	11.2 ± 0.8	10.7 ± 0.8	−0.5 ± 0.4	0.28	0.15	<0.01	0.15
Hand Grip Strength-Dominate (kg)	21.3 ± 0.1	22.9 ± 1.1	1.6 ± 0.1	0.05	21.1 ± 1.3	20.6 ± 1.6	−0.5 ± 1.1	0.67	0.46	0.44	0.14
**Self-Report Measures (VAS)**											
Fatigue (mm)	45.6 ± 6.0	26.9 ± 5.9	−18.8 ± 4.4	<0.01	46.7 ± 10.0	35.5 ± 9.7	−11.2 ± 5.4	0.07	0.64	<0.01	0.32
Quality of Life (mm)	73.7 ± 3.7	77.8 ± 3.5	4.1 ± 2.9	0.17	77.7 ± 5.0	84.2 ± 2.7	6.5 ± 3.3	0.08	0.35	0.04	0.62
Tiredness (mm)	44.8 ± 6.7	33.5 ± 5.4	−11.3 ± 7.1	0.13	37.2 ± 8.9	36.7 ± 8.8	−0.5 ± 0.9	0.58	0.82	0.28	0.32
Pain (mm)	32.1 ± 5.1	20.2 ± 4.4	−11.9 ± 5.5	0.04	13.2 ± 4.3	10.6 ± 3.5	−2.6 ± 3.5	0.47	0.03	0.10	0.28
**Cardiometabolic Health**											
HOMA-IR	6.1 ± 1.4	8.7 ± 2.6	2.6 ± 2.4	0.28	10.6 ± 4.2	7.7 ± 1.5	−4.8 ± 4.7	0.33	0.81	0.64	0.12
MetS	2.7 ± 0.4	2.4 ± 0.3	−0.3 ± 0.2	0.20	2.6 ± 0.6	2.9 ± 0.4	0.3 ± 0.3	0.39	0.74	0.97	0.15
Leptin (pg/mL)	34,148 ± 3940	35,170 ± 4616	1022 ± 2692	0.71	18,083 ± 3097	28,327 ± 6337	10,244 ± 5851	0.11	0.09	0.05	0.11
Adiponectin (ng/mL)	920.5 ± 4.3	894.8 ± 25.4	−25.7 ± 23.6	0.29	883.1 ± 19.7	911.3 ± 8.3	28.2 ± 22.8	0.25	0.62	0.95	0.16
Resistin (pg/mL)	16,941 ± 733	16,660 ± 597	−281 ± 965	0.69	15,334 ± 1669	15,488 ± 1236	155 ± 1286	0.91	0.27	0.93	0.75
**Inflammation and KYN Metabolism**											
hs-CRP (mg/L)	2.7 ± 0.6	2.8 ± 1.0	0.1 ± 0.6	0.85	2.9 ± 0.9	2.2 ± 0.5	−0.7 ± 0.9	0.47	0.86	0.58	0.44
KYN (pg/mL)	178.5 ± 15.6	112.3 ± 7.0	−66.2 ± 14.7	<0.01	103.5 ± 10.2	136.6 ± 19.3	33.1 ± 24.4	0.21	0.04	0.06	0.02
KYNA (pg/mL)	26.2 ± 2.5	29.5 ± 2.3	3.4 ± 2.1	0.13	29.2 ± 3.0	34.3 ± 2.9	5.0 ± 4.7	0.31	0.25	0.07	0.72
PGC−1α (pg/mL)	1073 ± 137	1477 ± 209	404 ± 152	0.02	856 ± 127	840 ± 129	−16 ± 124	0.90	0.09	0.08	0.05
KYN/KYNA Ratio	8.9 ± 1.6	4.2 ± 0.3	−4.8 ± 1.5	<0.01	3.6 ± 0.2	4.0 ± 0.4	0.4 ± 0.6	0.44	0.04	0.06	0.02

Unadjusted mean ± SEM; 6MWD: six-minute walk distance; TUG: timed up and go; VAS: visual analog score; HOMA-IR: homeostatic model assessment for insulin resistance; MetS: metabolic syndrome; hs−CRP: high-sensitivity C-reactive protein; KYN: kynurenine; KYNA: kynurenic acid; PGC-1α: peroxisome proliferator−activated receptor gamma coactivator 1-alpha. *p* < 0.05 differs from resistance training group at baseline.

## Data Availability

All data and material support our claims and comply with the field standards. The datasets generated during and/or analyzed during the current study are available from the corresponding author on reasonable request.

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
