# Peer review of "Kynurenine Metabolism as a Mechanism to Improve Fatigue and Physical Function in Postmenopausal Breast Cancer Survivors Following Resistance Training"

_jfmk, 2022, doi:10.3390/jfmk7020045_

Round 1
Reviewer 1 Report
This study deals with a very interesting topic, adding important information on the effects of RT on KYN metabolism and inflammation as a mechanism to promote changes in behavioral and physical function in postmenopausal BCS. In addition it supports the important opinion that exercise is an important non-pharmacological intervention for promoting psychological and functional health.
The design of the study is well structured, the methodology and statistical analyses are correct.
In addition, the results and conclusions are clear and exhaustive.
I have only some aspects need to be clarified.
2.2.1. Anthropometrics. I suggest to change the term ‘Anthropometrics’ with ‘Anthropometric measurements’
Line 218-219. Please specify why you have chosen to measure waist circumference “at the greatest anterior extension of the abdomen”. At this purpose, please consider “Waist Circumference and Waist-Hip Ratio Report of a WHO Expert Consultation, 2008”. As a consequence, please verify the validity of the cut-off (≥88 cm 152 for women and ≥102 cm for men) for your measurement (line 152).
The information “≥102 cm for men” is not necessary for the purpose of this study.
Line 215. “The women were majority Caucasian (60%)”. Please specify the ethnic composition of the sample.
Table 1 and lines 214-216. Have all the differences between RT group and CBCT group at baseline been considered? It is not so clear and it seems that some conditions (age, post active breast cancer (months), BMI,…) could be different.
Lines 220-222. “However, baseline pain was 1.4 times higher in the RT vs CBCT group (P=0.03) …..…….. except for concentrations of leptin and kynurenine were slightly higher in the RT vs. CBCT group (P’s <0.05)” I think that the baseline conditions needed to be similar between the two groups.
This could have influenced the results.
For these aspects, the experimental design and the results need clarification.
Author Response
Response to Reviewer 1 Comments
Point 1: I have only some aspects need to be clarified. 2.2.1. Anthropometrics. I suggest to change the term ‘Anthropometrics’ with ‘Anthropometric measurements’
Response 1: Thank you for this suggestion. We have changed section 2.2.1. to “Anthropometric Measurements.”
Point 2: Line 218-219. Please specify why you have chosen to measure waist circumference “at the greatest anterior extension of the abdomen”. At this purpose, please consider “Waist Circumference and Waist-Hip Ratio Report of a WHO Expert Consultation, 2008”.
Response 2: We thank the reviewer for requesting additional information on the technique we used to measure waist circumference. Currently, no consensus exists on the optimal protocol for measuring waist circumference among the five common methods used (e.g., WHO, NIH, MESA). For this study, we used the measurement protocol that the NIH provided to the Multi‐Ethnic Study of Atherosclerosis (MESA), which indicates that the waist measurement should be made at the level of the umbilicus or navel. This method is now cited.
Additionally, a 2019 study by Ostchega et al. compared the sensitivity and specificity for abdominal obesity between the Heart, Lung, and Blood Institute method (NHLBI- reference group) and the WHO and MESA methods and found that in women, sensitivity and specificity for abdominal obesity for WHO and MESA (Gulick- flexible vinyl tape) methods were both greater than 85%. Therefore, with WHO and MESA having similar sensitivity and specificity we felt confident in using the MESA protocol, the technique our research team had been previously trained on, thus confident the method would be performed accurately and consistently. To clarify, we added the following information to Section 2.2.1:
“Using the standardized protocol, maximal waist circumference was measured at the greatest anterior extension of the abdomen, usually at the level of the umbilicus, with a Gulick (flexible tape that does not stretch) tape and the participant standing upright and relaxed.”
Point 3: Please verify the validity of the cut-off (≥88 cm for women) for your measurement (line 152).
Response 3: A 2009 Joint Interim Statement of the International Diabetes Federation Task Force on Epidemiology and Prevention; National Heart, Lung, and Blood Institute; American Heart Association; World Heart Federation; International Atherosclerosis Society; and International Association for the Study of Obesity concluded that one of the risk factors for metabolic syndrome is elevated waist circumference with cut-off points specific to various populations and countries. In the United States, the American Heart Association National Heart, Lung, and Blood Institute (AHA/NHLBI (ATP III)) recommends waist circumference cut-off point of ≥88 cm for women. We have clarified and referenced this in Section 2.5.
Point 4: The information “≥102 cm for men” is not necessary for the purpose of this study.
Response 4: We agree and have removed the un-necessary information from the manuscript.
Point 5: Line 215. “The women were majority Caucasian (60%)”. Please specify the ethnic composition of the sample.
Response 5: We added the following specific ethnic composition to the manuscript: “Women were 60% Non-Hispanic White and 40% Non-Hispanic African American.” This information is also available in Table 1.
Point 6: Table 1 and lines 214-216. Have all the differences between RT group and CBCT group at baseline been considered? It is not so clear and it seems that some conditions (age, post active breast cancer (months), BMI,…) could be different.
Response 6: Several demographic characteristics were compared at baseline, including age, race, breast cancer latency, and BMI. Only age was found to be statistically different between group. We have added these baseline comparisons P-values in section 3.1 of the manuscript. Age was the only demographic found to be significantly different between groups; however, the difference (~6 years) is not considered clinically meaningful. When defining meaningful age ranges in the context of diseases that are governed by similar age-related processes, the latent Dirichlet allocation (LDA) clustering method defines 13 age clusters, including 50-73 years as one cluster (Geifman et al., 2013). The LDA is a widely accepted method and age ranges are grouped based on biomedical significance. These age ranges also reflect social, psychological, and/or biological process that co-occur in the same age group and thus describes important ages in the context of patient health. Based on LDA the mean age of the two groups is clustered within the same biologically relevant age ranges (50-73 years).
We also consulted with our biostatistician and based on the small sample size, along with mean ages clustered within the same biologically relevant age ranges, it was recommended that the analyses not be covaried for age. Although covarying can protect against chance imbalance and results in substantial increase in power when the covariates are highly prognostic (age is a well-documented predictor of physical and cardiometabolic functioning), covariate adjustment can lead to inflated type I error rates when there is a small sample size (Kahan et al., 2014). We now address this limitation in the discussion.
Point 7: Lines 220-222. “However, baseline pain was 1.4 times higher in the RT vs CBCT group (P=0.03) …..…….. except for concentrations of leptin and kynurenine were slightly higher in the RT vs. CBCT group (P’s <0.05)” I think that the baseline conditions needed to be similar between the two groups. This could have influenced the results. For these aspects, the experimental design and the results need clarification.
Response 7: We agree with the reviewer that future, powered studies are needed to allow for control of baseline differences that might have an effect on the outcome of interest. However, per consulting with our biostatistician, the 2-way ANOVA is the recommended statistical test with the current study design, as the ANOVA controls for baseline differences within the test. With the ANOVA model, variables of interest for each treatment group can be compared relative to their baseline.
References:
Geifman N, Cohen R, Rubin E. Redefining meaningful age groups in the context of disease. Age. 2013 Dec;35(6):2357-66. doi: 10.1007/s11357-013-9510-6. Epub 2013 Jan 27. PMID: 23354682; PMCID: PMC3825015
Kahan BC, Jairath V, Doré CJ, Morris TP. The risks and rewards of covariate adjustment in randomized trials: an assessment of 12 outcomes from 8 studies. Trials. 2014 Apr 23;15:139. PMID: 24755011; PMCID: PMC4022337
Reviewer 2 Report
This is a solid study regarding an important health issue. It is interesting that both intervention groups showed decreased fatigue, improved QoL, and increased leptin levels, while the RT and CBCT groups diverged on KYN and PGC-1alpha measures. Did the study account for activities outside the intervention? Is it possible that the psychological effects of CBCT led to increased daily activity or other lifestyle changes (e.g., diet) that could have improved these measures? If these data were not collected, this possibility should be acknowledged. It is possible that participants' smartphones have these daily measures (#steps, miles, etc.) archived prior to and during the intervention. Either way, the divergence in these effects vs the biochemical interaction effects should be addressed, if only speculatively.
Other issues
The units for the Visual Analog Scales is unclear (section 2.4.). Are the units "...ranging from 0-100 mm and higher..." really in mm for this scale? And what does the "and higher" mean?
For laboratory analyses, there is no mention of the blood collection procedure or the conditions under which this occurred. Section 2.5 just mentions that plasma was separated 48 hours after the last session. What volume was collected? Were subjects fasted (this is assumed based on the HOMA-IR methods)? How long? Line 325 in Discussion mentions a 24-48 hour window for blood collection post-RT/CBCT, but 2.5 mentions plasma separation at 48 hours post session. Was some of the blood frozen (24 hours post-session) and some (48 hours post-session) not? This should be clarified.
Why is there a separate statistical analysis for change for each intervention group when a 2-way ANOVA was conducted for group, time and the interaction effects? There is no untreated control group, so the within-treatment ANOVA seems inappropriate. The authors should consult a statistician.
For the Table, interpretation would be easier if the order of effects were Group, Time, Group X Time. Also, p values should either be expressed at the hundredths (e.g., 0.02) or using +, ++, and +++ to denote p<0.05, p<0.01, and p<0.001, respectively, using asterisks and Table notes.
Author Response
Response to Reviewer 2 Comments
Point 1: Did the study account for activities outside the intervention? Is it possible that the psychological effects of CBCT led to increased daily activity or other lifestyle changes (e.g., diet) that could have improved these measures? If these data were not collected, this possibility should be acknowledged. It is possible that participants' smartphones have these daily measures (#steps, miles, etc.) archived prior to and during the intervention. Either way, the divergence in these effects vs the biochemical interaction effects should be addressed, if only speculatively.
Response 1: We agree with the reviewer that assessment of these non-intervention changes in behavior would allow for further verification of the direct benefits of the RT and CBCT interventions. Unfortunately, these data were not collected at the time of the intervention and are not currently available. To address this issue, we have added the following to the limitations section: “Another limitation is that the study did not account for activities performed outside the intervention. Therefore, it is possible that the psychological effects of CBCT led to an increase in daily activity or other lifestyle changes (e.g., diet) which could have resulted in the improvement of fatigue and QoL.”
Point 2: The units for the Visual Analog Scales is unclear (section 2.4.). Are the units "...ranging from 0-100 mm and higher..." really in mm for this scale? And what does the "and higher" mean?
Response 2: The following information on Visual Analog Scales was added to section 2.4: “VAS consists of a 100 mm horizontal line with each end anchored with the extremes of the symptoms (i.e., left end “no pain” and right end “worst possible pain”). Participants mark the line indicating the amount of symptom they are feeling at the time. The score is determined by measuring the distance (mm) from the left end of the line to the participant’s mark. Scores for VAS range from 0-100 mm with a longer distance (mm) indicating a greater intensity of pain, tiredness, and fatigue and better QoL.”
Point 3: For laboratory analyses, there is no mention of the blood collection procedure or the conditions under which this occurred. Section 2.5 just mentions that plasma was separated 48 hours after the last session. What volume was collected? Were subjects fasted (this is assumed based on the HOMA-IR methods)? How long? Line 325 in Discussion mentions a 24-48 hour window for blood collection post-RT/CBCT, but 2.5 mentions plasma separation at 48 hours post session. Was some of the blood frozen (24 hours post-session) and some (48 hours post-session) not? This should be clarified.
Response 3: We have clarified the blood collection procedure by adding the following to section 2.5: “Twenty ml of EDTA plasma was drawn following a 12 hour fast and 24-48 hours after the last RT and CBCT session. Plasma was processed according to Standard Operating Procedures in Clinical Research. After collection, blood was mixed by inverting the tube 8-10 times, plasma was then immediately separated by centrifugation at 4°C for 10 minutes at 1100xg and then stored at -80°C until laboratory analyses.”
Point 4: Why is there a separate statistical analysis for change for each intervention group when a 2-way ANOVA was conducted for group, time and the interaction effects? There is no untreated control group, so the within-treatment ANOVA seems inappropriate. The authors should consult a statistician. Response 4: We thank the reviewer for requesting this clarification. We have reconsulted with our biostatistician regarding the appropriate statistical analyses for the current project. The current method of a 2-way ANOVA continued to be recommended as the ANOVA allows for comparisons of differences across time between groups. The ANOVA allows for testing of mean differences between the groups, even if both interventions are active (non-control) interventions. Because traditional post-hoc testing in a repeated measures framework does not fully account for intra-individual variability, our biostatistician recommended that within group changes be explored using paired t-tests. This has been updated in the methods.
Point 5: For the Table, interpretation would be easier if the order of effects were Group, Time, Group X Time. Also, p values should either be expressed at the hundredths (e.g., 0.02) or using +, ++, and +++ to denote p<0.05, p<0.01, and p<0.001, respectively, using asterisks and Table notes.
Response 5: We thank the reviewer for providing format suggestions to the table. We have updated the order of effect in the table to Group, Time, Group X Time and the p values are now expressed at the hundredths.
Reviewer 3 Report
The study addresses a very interesting and, above all, necessary topic. The presented study is a pilot study evaluating the effect of resistance training on a number of parameters.
Theoretical data are processed in a satisfactory manner, methodological procedures are described, clearly, transparently, as well as the results are written in a suitable manner and documented by a clear, albeit comprehensive, table. The discussion is adequate given the results and topics addressed in the study.
In the overall evaluation, I consider the manuscript to be well prepared. However, in my opinion, I consider a very small group of probands to be one of the major shortcomings, as well as an uneven distribution of the research and control groups (distribution 2: 1).
Despite the fact that this is a pilot study, this fact could have significantly affected the results obtained. Despite this shortcoming, the study brings certain results that can be used in practice and, above all, can be the basis for further studies.
Author Response
Response to Reviewer 3 Comments
Point 1: In the overall evaluation, I consider the manuscript to be well prepared. However, in my opinion, I consider a very small group of probands to be one of the major shortcomings, as well as an uneven distribution of the research and control groups (distribution 2: 1).
Response 1: We agree with the reviewer that this pilot is limited by the small sample size, which we mention as a limitation. Barring the fixed research budget and time limitations of this pilot and given that the effects of resistance training were our primary intervention of interest, we opted to recruit 2:1 because more statistical power could be obtained by allocating more participants to the treatment group of primary interest (RT) while allowing us to maximize the budget and time restraints. The unequal randomization of 2:1 allowed us to gain more information about the intervention. Despite the small sample size, we believe that these data add value to the current scientific literature by providing preliminary support for the benefit of RT and CBCT on fatigue reduction in chronic breast cancer survivors, as well as providing preliminary data to adequately power future clinical trials.
Round 2
Reviewer 1 Report
The Authors have taken into account all my suggestions; thus in this form the manuscript is acceptable for publication.